# Microbially Mediated Rubber Recycling to Facilitate the Valorization of Scrap Tires

**DOI:** 10.3390/polym16071017

**Published:** 2024-04-08

**Authors:** Sk Faisal Kabir, Skanda Vishnu Sundar, Aide Robles, Evelyn M. Miranda, Anca G. Delgado, Elham H. Fini

**Affiliations:** 1School of Sustainable Engineering and the Built Environment, Arizona State University, 660 S College Ave, Tempe, AZ 85281, USA; 2Center for Research and Education in Advanced Transportation Engineering Systems (CREATEs), Rowan University, South Jersey Technology Park, 107 Gilbreth Parkway, Mullica Hill, NJ 08062, USA; 3Biodesign Swette Center for Environmental Biotechnology, Arizona State University, 1001 S McAllister Ave, Tempe, AZ 85281, USAemmirand@asu.edu (E.M.M.); 4School for Engineering of Matter, Transport & Energy, Arizona State University, 501 E Tyler Mall, Tempe, AZ 85287, USA

**Keywords:** sustainability, low-carbon economy, bitumen compatibility, desulfurization, rubberized bitumen, microbial mediation

## Abstract

The recycling of scrap tire rubber requires high levels of energy, which poses challenges to its proper valorization. The application of rubber in construction requires significant mechanical and/or chemical treatment of scrap rubber to compatiblize it with the surrounding matrix. These methods are energy-consuming and costly and may lead to environmental concerns associated with chemical leachates. Furthermore, recent methods usually call for single-size rubber particles or a narrow rubber particle size distribution; this, in turn, adds to the pre-processing cost. Here, we used microbial etching (e.g., microbial metabolism) to modify the surface of rubber particles of varying sizes. Specifically, we subjected rubber particles with diameters of 1.18 mm and 0.6 mm to incubation in flask bioreactors containing a mineral medium with thiosulfate and acetate and inoculated them with a microbial culture from waste-activated sludge. The near-stoichiometric oxidation of thiosulfate to sulfate was observed in the bioreactors. Most notably, two of the most potent rubber-degrading bacteria (*Gordonia* and *Nocardia*) were found to be significantly enriched in the medium. In the absence of added thiosulfate in the medium, sulfate production, likely from the desulfurization of the rubber, was also observed. Microbial etching increased the surface polarity of rubber particles, enhancing their interactions with bitumen. This was evidenced by an 82% reduction in rubber–bitumen separation when 1.18 mm microbially etched rubber was used. The study outcomes provide supporting evidence for a rubber recycling method that is environmentally friendly and has a low cost, promoting pavement sustainability and resource conservation.

## 1. Introduction

Rubberized asphalt binders have been shown to be superior in terms of durability, service life, and reduced noise compared to neat binders [1,2,3]. However, sulfur-crosslinked rubber, known as vulcanized rubber, does not disperse well in asphalt binders [4,5]. Since recycled crumb rubber is rich in sulfur-crosslinked rubber particles, it mainly works as an elastic particulate and can give rise to the segregation of rubber and asphalt [4,6]. The swelling of rubber particles in the asphalt binder and the density difference between rubber and bitumen also contribute to the separation of rubber from bitumen [4,7].

To improve the interaction between crumb rubber and asphalt, treatments using UV [8], plasma [9,10], microwaves [4,6,11], and bio-oils [12] or chemical grafting [13,14] have been explored. In addition to physical and chemical techniques, microbial methods have also been utilized to etch or desulfurize the surface of rubber [15,16,17]. Microorganisms like *Acidithiobacillus* sp. and *Sphingomonas* sp. can couple the oxidation of sulfur with the reduction of oxygen (aerobic process) as an energy-yielding process [18,19,20,21,22,23]. In a study using *Acidithiobacillus* to treat crumb rubber, microbial desulfurization was 22% more effective than a chemical treatment by utilizing the Neospagnol T-20 method [22]. It has been reported that by using *Sphingomonas* sp., S–C and S–S bond scission occurred in the desulfurized rubber [24]. A significant amount of desulfurization (62.5%) has been reported in a study with *Alicyclobacillus* sp. [25].

The size of rubber particles and the dosage are two critical factors that affect the modification of rubberized asphalt to a great extent [26]. Qian and his research group studied the influence of rubber-particle size in crumb-rubber-modified asphalt. They studied four size ranges: 0.15–0.30 mm, 0.30–0.45 mm, 0.45–0.60 mm, and 0.60–0.75 mm. According to their observations, asphalt modified with larger crumb rubber particles had a lower fatigue parameter (G*sinδ), indicating a higher resistance to fatigue cracking [27]. They also reported that larger-particle-sized crumb rubber can also resist rutting [27]. Lei and his research group completed a size variability study on the hybrid method of combining microwave heat treatment with bio-oil. For this purpose, they chose crumb-rubber particles with sizes of 0.85 mm, 0.38 mm, 0.18 mm, and 0.15 mm. According to their findings, with the increased size of rubber particles, the fatigue life was increased. They reported that the asphalt binder with the 0.85 mm rubber size had the highest resistance to permanent deformation [26].

In a previous study, we showed that microbial etching with the particle size of ≤0.25 mm diameter was beneficial for the modification of the asphalt binder [15]. Specifically, we demonstrated that phase separation decreased by 68% along with a 27% decrease in moisture diffusion compared to conventional similar-sized particles.

Building on our past effort, this current paper studies whether microbial treatment can accommodate rubber particles of various sizes. This, in turn, can significantly reduce pre-processing time, costs, and environmental impacts. Key indicators will include the enrichment of rubber-degrading bacteria, Gordonia and Nocardia, and improved rubber–bitumen interactions. 

## 2. Materials and Methods

### 2.1. Materials

Crumb rubber was collected from Crumb Rubber Manufacturers, Mesa, AZ, USA. The rubbers were sieved using #16 and #30 size meshes to obtain particles that passed through 1.18 mm and 0.6 mm sieve opening. The virgin binder used in this study was PG 64-22 (Table 1), supplied by Hollyfrontier, Glendale, AZ, USA.

### 2.2. Experiment Setup for the Microbial Etching of Crumb Rubber

Microbial etching experiments were performed in bioreactors consisting of 500 mL Erlenmeyer flasks (Corning, NY, USA) with 50 g of crumb rubber. The conditions tested are shown in Table 2. The mineral medium was prepared as described in Kabir et al. [15]. Briefly, 10 mM potassium phosphate and 25 mM sodium acetate were added to the medium and the pH was adjusted to ~7.5. The inoculum was 10 mL of waste-activated sludge (WAS) collected from the Greenfield Wastewater Treatment Plant, Gilbert, AZ, USA. Conditions labelled “CR30” contained crumb rubber with a diameter of 0.6 mm, while conditions labelled “CR16” contained crumb rubber with a diameter of 1.18 mm (Table 2).

The conditions inoculated with waste-activated sludge (CR30-2, CR30-3, CR16-2, and SC) were set up in triplicate, while the rest were set up in duplicate (Table 2). The flasks were incubated at 30 °C and shaken at 150 rpm for 41 days (end of experiment). Sodium acetate (25 mM) was re-added on day 20 in conditions CR30-1, CR30-2, CR16-1, and CR16-2. The pH was adjusted to 7.5 with a solution of 2 M HCl on days 3, 30, and 36 for conditions CR30-1, CR30-2, CR16-1, and CR16-2.

Liquid samples from bioreactors (~10 mL) were collected at least once per week and stored at −20 °C for further analyses. A mineral medium (without sulfate and thiosulfate) was added to restore the experiments to their original volume after each sampling event.

### 2.3. Chemical Analysis

The pH was monitored using a Thermo Scientific Orion Versa Star Pro meter equipped with an Orion 8157BNUMD ROSS Ultra pH/ATC Triode (Thermo Fisher Scientific, Waltham, MA, USA). A 3-point pH calibration was performed using Orion ROSS 4.01, 7.00, and 10.01 pH buffers (Thermo Fisher Scientific, Waltham, MA, USA). The sulfate and thiosulfate concentrations were quantified from samples filtered through 0.45 µm Nylon-66 syringe filters (Membrane Technologies, Harrisburg, PA, USA) using a Metrohm 930 Compact IC Flex equipped with a Metrosep A Supp 5–150/4.0 column and a conductivity detector (Metrohm, Riverview, FL, USA) [36,37]. The eluent used was a solution of 3.2 mM carbonate and 1 mM bicarbonate at a flow rate of 0.7 mL min^−1^. The calibrations were performed using Custom Anion Mix: 3 (Cat. No. REAIC1035, Metrohm, Riverview, FL, USA) in a range of 0.1–50 mg L^−1^.

Acetate concentrations were quantified from filtered samples using a high-performance liquid chromatograph (HPLC, Shimadzu LC-20AT Columbia, MD, USA) equipped with a photodiode array detector at 210 nm, a refractive index detector, and an Aminex HPX-87H column (BioRad, Hercules, CA, USA) using the methodology previously described [38,39]. A 5-point calibration was carried out using a volatile acid standard solution (AccuStandard, New Haven, CT, USA) in a range of 2.5–10 mM. The detection limit of the analytes was ≤0.4 mM.

### 2.4. Microbial Community DNA Amplicon Sequencing

Chromosomal DNA was extracted from waste-activated sludge (the inoculum), the rubber on day 0, the rubber on day 41 (end of experiment), and liquid samples on day 41 (end of experiment). All DNA extraction from the rubber samples (250 mg each) were accomplished by using a DNeasy Powersoil Pro kit (Qiagen, Germantown, MD, USA) following the manufacturer’s protocol. For DNA extraction from liquid, 1 mL of each sample was pretreated with an enzyme lysis buffer. The enzyme lysis buffer contained 20 mM Tris·HCl, 2 mM EDTA, 250 μg mL^−1^ achromopeptidase, and 20 mg mL^−1^ lysozyme. A Qiagen DNeasy Blood and Tissue kit (MO BIO Laboratories Inc., Carlsbad, CA, USA) was then used for DNA extraction from the WAS and liquid samples following the protocol for Gram-positive bacteria. Once extracted, the DNA yield and purity were determined with a Nanodrop 1000 spectrophotometer (Thermo Scientific, Waltham, MA, USA) and sent for sequencing to the Center for Fundamental and Applied Microbiomics at the Arizona State University KED Genomics Core Facility (Tempe, AZ, USA). 

Amplicon sequencing targeting the V4-hyper-variable region of the 16S rRNA gene was performed using the following primers from the Earth Microbiome Project: 515F (5′-GTGCCAGCMGCCGCGGTAA-3′) and 806R (5′-GGACTACHVGGGTWTCTAAT-3′) [40,41]. The amplicon sequence variants (ASVs) were imported into the QIIME2 software (QIIME2, v. 2021.4.0) [42] in the Casava 1.8 paired-end demultiplexed format. The sequences were demultiplexed and the DADA2 plug-in was used for sequence quality control [43], where the sequences were truncated at 232 bases. The resulting ASVs were assigned taxonomy in reference to the SILVA v. 138 database [44,45]. All forward- and reverse-sequence files from this study were deposited into the National Center for Biotechnology Information (NCBI) database under BioProject PRJNA1010810 and SRA accessions SRR25833191-SRR25833204.

### 2.5. Physical Analysis

Crumb rubber samples for physical analysis were collected from the flasks at the end of the experiment on day 41. The liquid from the crumb rubber that was microbially etched was first discarded and then the rubber was rinsed twice with acetone and then twice with DI water.

### 2.6. Modification of Bitumen

The supplied binder was separately modified by each of the four types of microbially mediated rubber to obtain a rubberized binder. Then, 15% of the desulfurized rubber was introduced to PG 64-22 by the weight of the base binder. A bench-top high-shear mixer was used at 3000 rpm for 30 min at about 165 °C to blend, validated by previously published work [15]. 

### 2.7. Fourier Transform Infrared Spectroscopy (FT-IR)

Attenuated Total Reflectance–Fourier Transform Infrared Spectroscopy (ATR-FTIR) was performed using a Bruker FT-IR instrument. This chemical characterization was carried out to observe any chemical bond change in the microbially mediated rubber. A mid-infrared range diamond ATR was used to detect changes in the 500 to 540 cm^−1^ range. A range of 4000 cm^−1^ to 400 cm^−1^ wave numbers was selected. Rubber samples were washed for five minutes using acetone to remove any treatment solution residue and to sterilize samples prior to FT-IR analysis. All the spectra were analyzed in Origin Pro 2018 Software.

### 2.8. Dynamic Shear Rheometer

Rheological characterization for this study is critical as there is a need to distinguish between these four types of rubber-modified asphalt. An Anton Paar Dynamic Shear Rheometer was used to collect data for each specimen for the phase separation study. For this test, the complex modulus (G*) and phase angle (δ) were calculated following ASTM D7173 [46]. The DSR was also used to calculate G*/sinδ and G*sinδ at 64 °C and 46 °C, respectively. In addition, the percentage recovery (*R*) of the binders was measured by performing the multiple-stress creep and recovery (MSCR) test at 58 °C following AASHTO M332. These oscillation tests were performed at ten rad/s to represent traffic conditions with a speed of 90 km/h [47].

### 2.9. Phase Separation Analysis

In the wet process of preparing rubberized asphalt, the phase separation of rubber particles from the asphalt binder is an issue that needs attention. Separation tendency was determined by heating the samples at 163 °C to make them sufficiently fluid. The fluid binder was then poured into aluminum tubes. It is important to ensure that the tube tops are sealed, as air may enter and oxidize the specimen. Then, the tubes were placed vertically in a sample-holder rack. Tubes were then placed inside an oven at 163 °C for 48 h. After 48 h, the rack was placed in a freezer for 4 h at −18 °C. Upon cooling, the tubes were taken out and cut into three equal segments [46]. The middle section was discarded, and the bottom and top sections were stored for tests with the dynamic shear rheometer at 58 °C. After calculating the complex modulus and phase angle, the segregation index (SI) was calculated according to Equation (1) [48].
(1)$$\mathrm{SI}=\frac{\left(\frac{G^{*}}{sin\delta }\right)max-\left(\frac{G^{*}}{sin\delta }\right)avg}{{(\frac{G}{sin\delta }}^{*}{)}_{avg}}$$
where G* = complex shear modulus and δ = phase angle.

### 2.10. Low-Temperature Cracking 

We then evaluated the crumb-rubber-modified binders to observe their low-temperature response. A bending beam rheometer (BBR) was used to evaluate low-temperature properties. A three-point bending test of a binder beam with a fixed length, width, and height performed under a cold bath of ethanol [49] measured the flexural creep stiffness (S) and stress relaxation capacity (m-value) by applying a load of 980 ± 50 mN for the duration of 240 s at the midpoint of the beam. The beam deflection (d) was measured at the center with respect to loading time using a linear variable differential transducer (LVDT) following the Superpave™ specification [50]. The stress-relaxation value determines the bitumen’s ability to prevent thermal cracking triggered by a sudden drop in temperature in a cold-climate environment. A testing temperature of −12 °C was selected for this experiment. The stiffness was calculated from the deformation data during the loading period, using Equation (2):(2)$$S(t)=\frac{PL^{3}}{4bh^{3}\partial (t)}$$
where:

P is the applied constant load (100 g or 0.98 N); 

L is the distance between beam supports (102 mm); 

b is the beam width (12.5 mm); 

h is the beam thickness (6.25 mm); 

S(t) is the bitumen stiffness at a specific time, MPa; 

∂(t) is the deflection at a specific time, mm.

## 3. Results and Discussion

Sulfate and thiosulfate concentrations were measured at regular intervals for the 41 days of the experiment. This was undertaken to monitor the rates and extent of microbial desulfurization. Figure 1 shows the time-course thiosulfate and sulfate concentrations in the flasks containing 50 g crumb rubber (CR) for each experimental condition. As seen in Figure 1, sulfate concentrations substantially increased during incubation for CR16-2, CR30-2, and SC because of the microbial conversion of thiosulfate to sulfate. However, compared to the sulfate release in some conditions, CR30-1 and CR16-1, not supplied thiosulfate and sulfate, had very little sulfate release. Initially, adding Thiosulfate increased the concentration of sulfate, but no change in sulfate was observed after day 7 during the incubation period. 

The sulfate release rates were slightly higher in the flasks with 50 g CR16-2 + WAS, compared to CR30-2. After 40 days of incubation, the highest final sulfate concentrations in CR16-2, sulfate, and CR30-2 were not significantly different; ~642 mg/L was observed in CR16-2 and ~553 mg/L in CR30-2 sulfate.

### 3.1. Microbial Community Analysis

Microbial abundance was determined in materials (i.e., WAS and rubber) and in bioreactors at the end of incubation (day 41) to identify key microorganisms and explore possible microbial pathways. Several genera such as *Brevundimonas*, *Methylobacterium*, *Gordonia*, and *Nocardia* containing rubber oxygenase A (RoxA) and rubber oxygenase B (RoxB), enzymes able to catalyze the oxidative cleavage of latex, were enriched in conditions CR16-2, CR30-2, and CR30-3, (Figure 2) [51,52]. Most notably, conditions inoculated with WAS and supplied with sulfate and thiosulfate showed an enrichment of *Gordonia* and *Nocardia*, two of the most potent rubber-degrading bacteria, from 0.03% and 0.08% to upwards of 3.97% and 1.71% in CR16-2 and CR30-2, respectively (Figure 2) [53,54]. Additionally, *Ketobacteria* and *Gammaproteobacteria*, known to contain RoxB, may have played a key role in degradation [55,56]. The relative abundance of *Ketobacteria* increased to 10% and 14% in conditions CR16-2 and CR30-2, respectively, while *Gammaproteobacteria* remained above 4% in conditions with rubber and WAS (Figure 2). *Alphaproteobacteria* with the Sox enzyme system, which allows bacteria to directly oxidize thiosulfate to sulfate, became dominant in conditions inoculated with WAS and supplied sulfate and thiosulfate (i.e., CR16-2 and CR30-2) [57]. The microbial community data are highly indicative of etching of the rubber particles, as many microorganisms known to play a key role in desulfurization were not only identified, but enriched under the various conditions tested.

### 3.2. Chemical Analysis

ATR-FTIR spectra measured for CR16-1, CR16-2, CR30-1, and CR30-2 samples in this study are shown in Figure 3. The spectra were normalized to the peak at 2800 cm^−1^. Figure 3a shows some greater changes during the microbial incubation for CR16-1 and CR30-1. The change is more evident in the 1040 cm^−1^ region of CR30-1. A similar change was evidenced in previous work on the surface activation of rubber [4]. However, any change in the weak disulfide bond region (500–540 cm^−1^) is noteworthy for these two samples as it indicates a breakage of bonds and release of sulfur. 

In Figure 3b, where the spectra of CR16-2 and CR30-2 are shown, the changes are more noticeable. The peak in the 1540 cm^−1^ region decreased for CR30-2. Most importantly, the weak disulfide bond seems to have been reduced for both cases compared to CR16-1 and CR30-1, providing a hint that partial desulfurization may have occurred.

### 3.3. Phase Separation

Phase separation of the microbially mediated rubbers was evaluated by a cigar tube test, following the standard ASTM D7173 [46]. Figure 4 shows segregation indexes calculated for each type of rubber-modified asphalt binder. It can be seen that for both rubber scenarios, rubber–bitumen phase separation has improved. After microbial treatment, the CR16-2 size rubber phase separation had decreased by almost 82%, whereas the CR30-2-size rubber had an approximately 55% decrease in phase separation. This is consistent with our previous finding, that microbial activity reduced phase separation in CR60-size rubber particles [15]. The phase separation of rubber from the asphalt binder is a major challenge in the widespread implementation of rubberized bitumen [58,59]. However, it can also be inferred from this test that adding nutrients nourishes the microbial community and allows it to thrive, and consequently, we can see partial desulfurization taking place. Breakage of the crosslinked sulfur bonds can reduce the amount of agitation required for crumb rubber that has not been desulfurized. Due to reduced viscosity [15], microbial mediation can allow a greater percentage of crumb rubber to be used in rubberized asphalt without segregation. It has been reported that currently, crumb rubber is added at up to 20% by the weight of the base asphalt binder in the wet process [59].

### 3.4. Rheological Analysis 

To better evaluate the effect of the microbial desulfurization of rubber on the asphalt binder, the G* and δ at 10 rad/s were used to calculate G*/sinδ at 64 °C and G*sinδ at 46 °C, since these two parameters are used to relate the rutting and fatigue characteristics of pavements, respectively. Figure 5a shows that the G*/sinδ values dropped a significant 82% after microbial etching for larger-particle-size rubbers. Interestingly enough, there was no significant increase in the G*/sinδ value for the two types of microbially treated #30 modified rubberized asphalt. Therefore, it is possible that for smaller sized rubber particles, the desulfurization will happen and then again create a network by bonding with the asphalt binder using the excess free sulfur. 

A similar trend is seen when G*sinδ at 46 °C was evaluated, as shown in Figure 5b. For resisting fatigue, a binder that is too soft or too stiff is undesirable. The CR16-2-modified binder had an approximate 69% decrease in G*sinδ. As seen in previous rubber-particle-size studies, the asphalt binder modified with larger-particle-sized rubber is more resistant to fatigue [26,27]; we see a similar trend here as well. In contrast, there is no significant difference in G*sinδ value for CR30-1- and CR30-2-modified binders. Indeed, microbial activity and high-shear mixing caused a drastic change in the CR16 samples, as shown by the bar charts in Figure 5.

Next, we evaluated the rubberized asphalt binder samples in a multiple-stress creep recovery (MSCR) test. The goal was to evaluate how much elasticity the microbially mediated rubber can provide to the asphalt binder. Figure 6 shows the percentage recovery (R) at 0.1 kPa and a traffic-simulated stress of 3.2 kPa at 58 °C. CR16-2 samples had only a 24% recovery at 3.2 kPa, which means that this rubberized binder became softer and never regained its elasticity; most probably, CR16-2 underwent chain scission and lost its elasticity, resulting in a drastic 60% decrease in percentage recovery. But for CR30-2, the opposite trend was noticed, and recovery increased from 53% to 57%. A slight increase in percentage recovery was also noticed in our previous experiment with CR60-size crumb rubber [15]. We can observe that a percentage recovery between 0.1 kPa and 3.2 kPa (known as R_diff_) was influenced by a larger particle size (1.18 mm).

### 3.5. Low-Temperature Performance

Figure 7 shows the low-temperature properties of the rubberized asphalt binders in terms of stiffness and stress relaxation ‘m’ value. The test was performed at −12 °C. For a polymer to have effective low-temperature properties, it is desirable to have lower stiffness and higher m-value. It can be seen that CR16-2 had a noticeable reduction in stiffness compared to the CR16-1 modified sample. For this particular case, stiffness decreased nearly 32% and the “m” value increased by approximately 31%. The other three samples had very similar stiffness with very similar stress relaxation values. Most probably, the effect of treatment on larger particle sizes is more evident than on the smaller ones, as can be seen in the increase in stress relaxation value. 

### 3.6. Industry Implications and Carbon Management Perspective

Considering the significant emphasis on net zero carbon emission roads, the study outcomes are expected to provide insights into crumb-rubber-modified asphalt. It should be noted that the CO_2_-equivalent values for the crumb-rubber-modified asphalt binder are reported to be 0.745 Kg CO_2_/ton, which is still lower than asphalt modified with PPA, SBS, and even non-modified asphalt, which had values of 0.786, 0.918, and 0.766, respectively [60]. The carbon content of treated crumb rubber comes from the recycling steps of waste tires as well as post-treatments. While there have been treatments [61] for crumb rubber particles to enhance their interactions with asphalt binders, these methods are costly, time-consuming, and require customized plant equipment. The microbial treatment used in this study has a lower carbon footprint and enhances rubber–asphalt interactions by etching the surface of rubber particles to alter its surface chemistry for better binding with asphalt. In terms of carbon counting, if no emission occurred during the recycling process, the net CO_2_ equivalent would be equal to the total carbon sequestered in waste rubber. However, the processes of scrap tire collection, transportation, storage, grinding, washing, and treatment each have their own associated CO_2_ emissions which need to be accounted for. The carbon emissions for recycling scrap tires are reported to be 124 kg CO_2_ equivalent per ton, the majority of which is from processing and transport [62]. Added to that is the carbon associated with the treatment applied to rubber crumbs before they are introduced to asphalt. Considering the need for significantly higher energy and chemicals needed for both physisorption and/or chemical treatments [61], the CO_2_ emissions of microbially treated rubber are expected to be significantly lower than those of conventional rubber. 

## 4. Conclusions

This paper investigated the efficacy of a microbially mediated desulfurization process on scrap-tire rubber of different particle sizes. The successful desulfurization of rubber particles leads to the devulcanization of the rubber polymer to make it more available for use in industrial and construction applications. To study the method’s efficacy, we modified the surface of two different rubber particle sizes of 1.18 mm and 0.6 mm by incubating them in a medium with microorganisms from WAS. The study results showed that the microbial treatment method was effective regardless of particle size. The following conclusions were drawn from this study:
The surface chemistry of rubber particles changed after microbial treatment, as shown by the reduction in peak intensity at 500–540 cm^−1^ in FTIR spectra.The extent of rubber–bitumen separation was greatly reduced after microbial treatment, as deduced from the 55 to 82% reductions in the Segregation Index in the studied particle sizes (0.6 and 1.18 mm).Bitumen containing treated rubber (CR16-2) showed a reduction in percentage recovery, indicating reduced elasticity of rubber particles; this can be attributed to the breakage of the sulfur crosslinks of the rubber after microbial treatment. This was evidenced by the high release of sulfur in the process.The enrichment of the known rubber degraders *Gordonia* and *Nocardia*, and other microorganisms containing rubber oxygenases, supported chemical and physical evidence of microbial treatment.Bitumen containing treated rubber (CR16-2) showed an increased resistance to fatigue cracking, indicating the contribution of rubber polymers to bitumen’s mechanical properties.

Based on the study results, it can be concluded that the microbially mediated desulfurization process is a promising alternative method to devulcanize scrap-tire rubber particles. Future study is recommended to optimize the process in terms of temperature and growth media as well as the microbial community of the sludge.

## Figures and Tables

**Figure 1 polymers-16-01017-f001:**
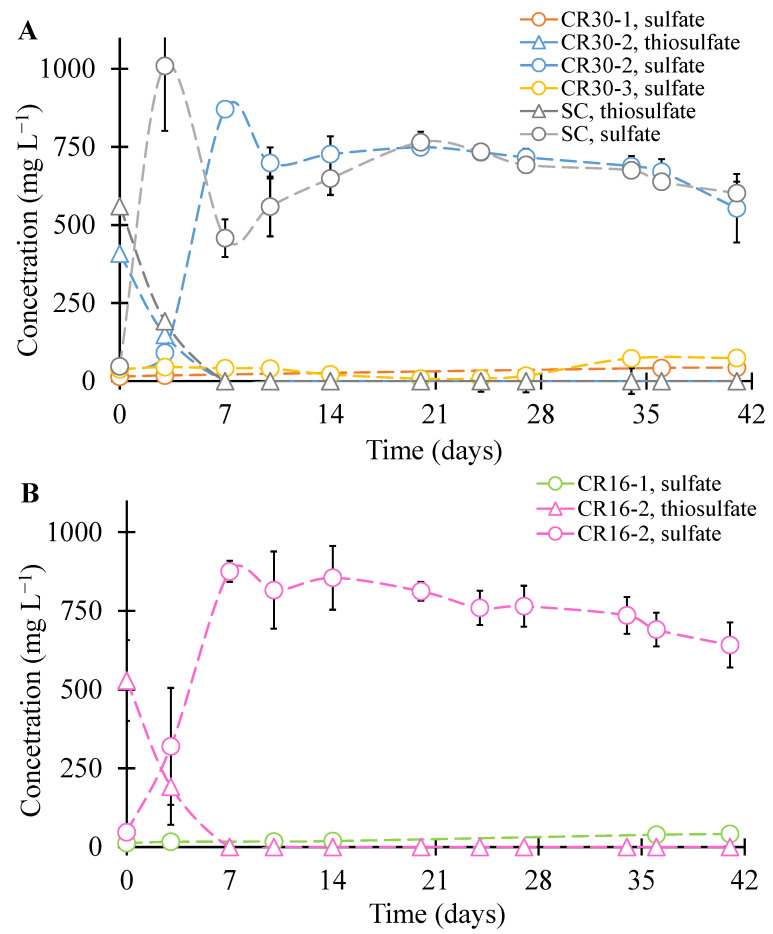
Thiosulfate and sulfate concentrations in flask bioreactors with (**A**) crumb rubber 30 (CR-30) and (**B**) CR-16 during incubation. SC = sludge control (no crumb rubber). The data are averages with SD of duplicate or triplicate bioreactors.

**Figure 2 polymers-16-01017-f002:**
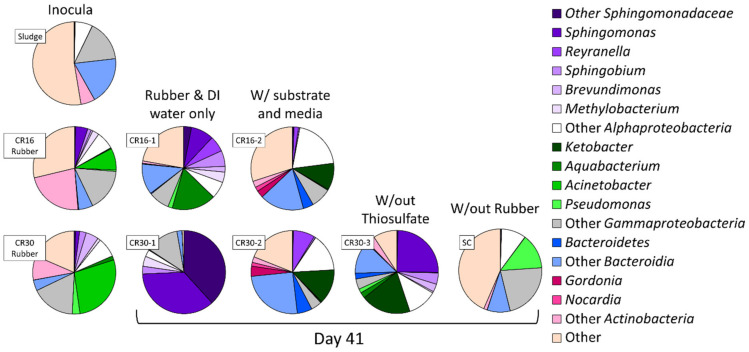
Microbial community composition of the inoculate at day 0 and in the bioreactors at the end of incubation (day 41).

**Figure 3 polymers-16-01017-f003:**
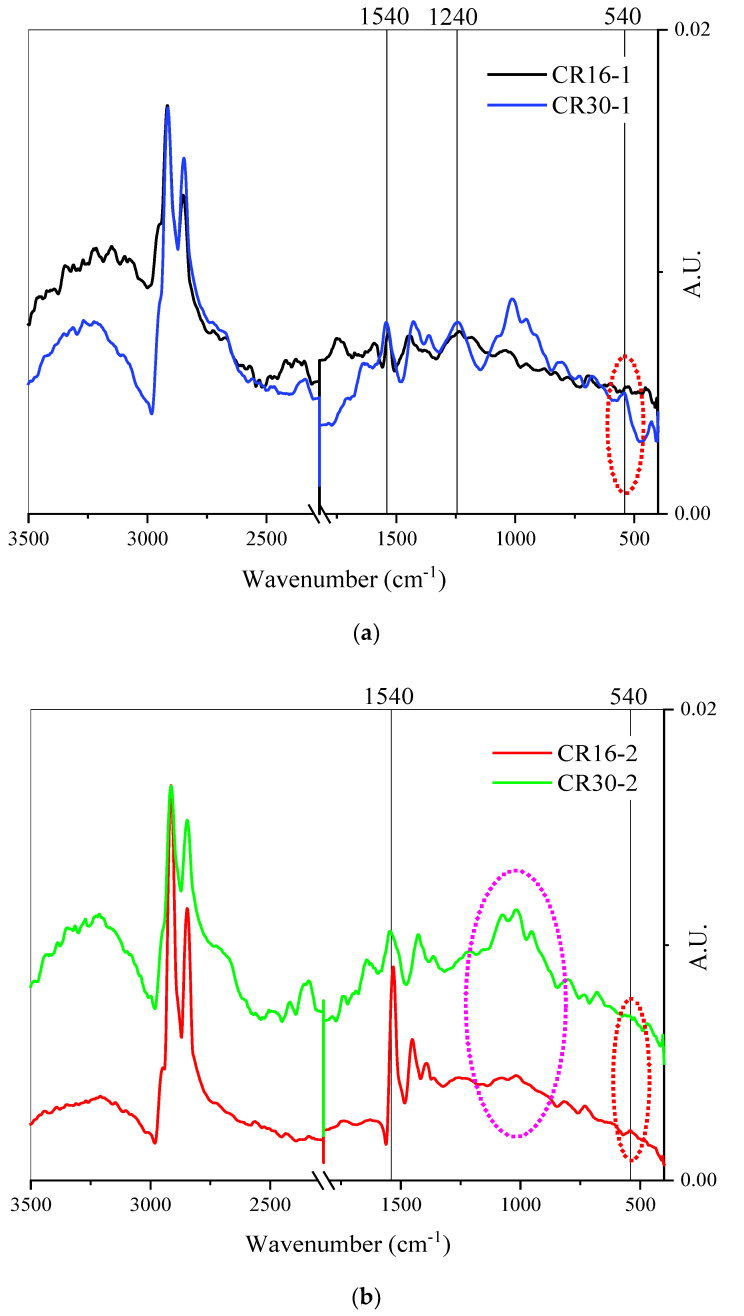
FT-IR spectra of both crumb rubber sizes after acetone washing. FT-IR spectra in both cases draw a comparison between (**a**) CR16-1 and CR30-1 and (**b**) CR16-2 and CR30-2.

**Figure 4 polymers-16-01017-f004:**
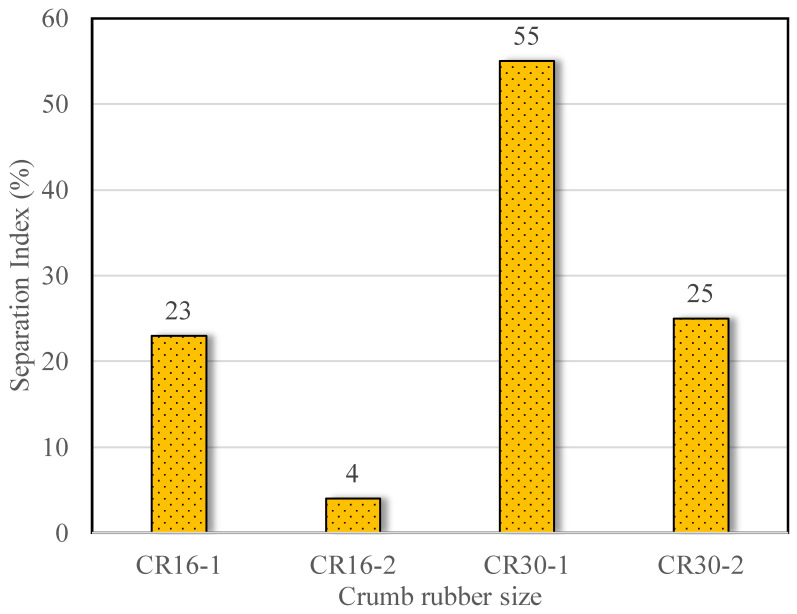
Cigar tube test results to study the phase separation phenomenon of the rubberized asphalt binder.

**Figure 5 polymers-16-01017-f005:**
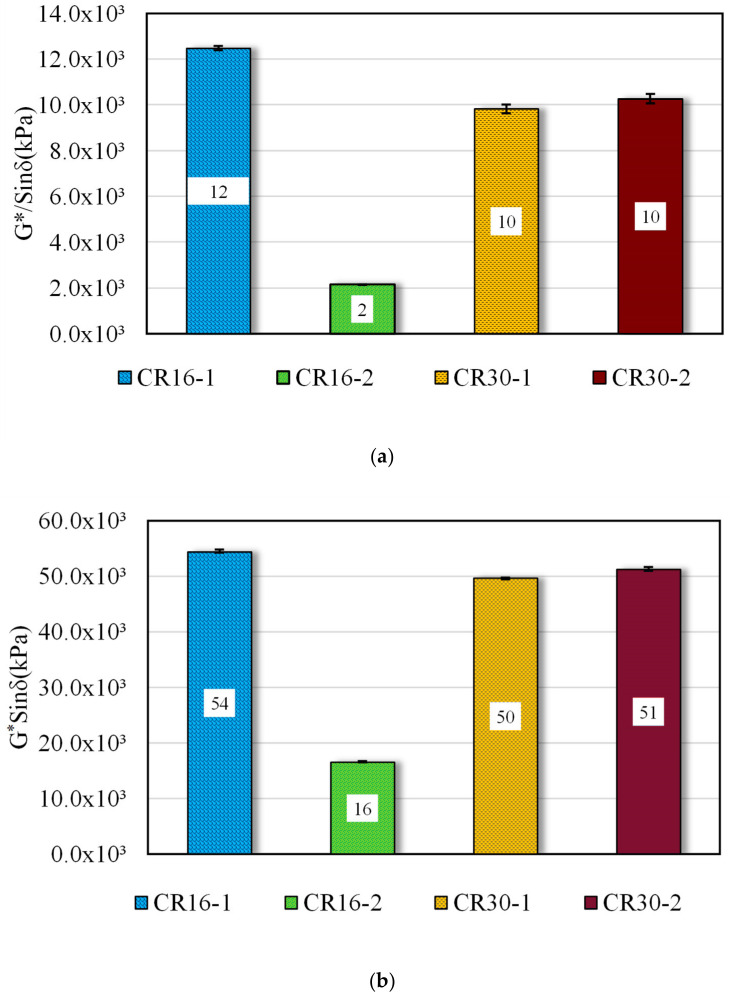
(**a**) G*/sinδ and (**b**) G*sinδ values for microbially desulfurized rubberized asphalt binders.

**Figure 6 polymers-16-01017-f006:**
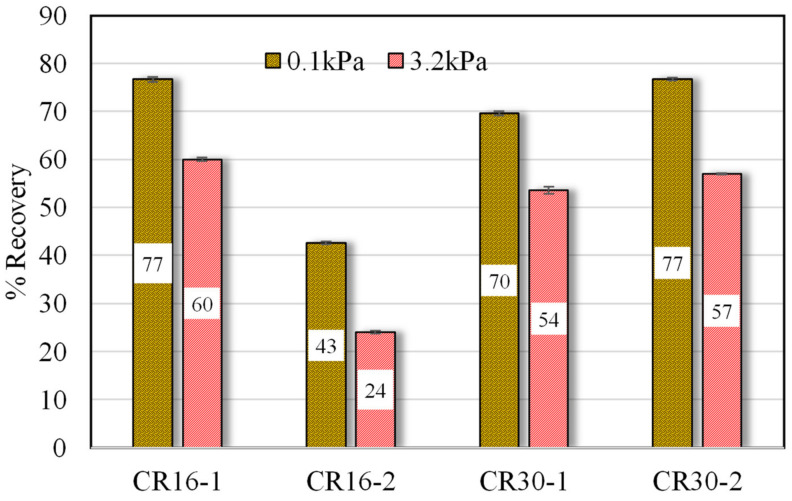
Percentage recovery of the accumulated strain from MSCR test results for rubberized binders.

**Figure 7 polymers-16-01017-f007:**
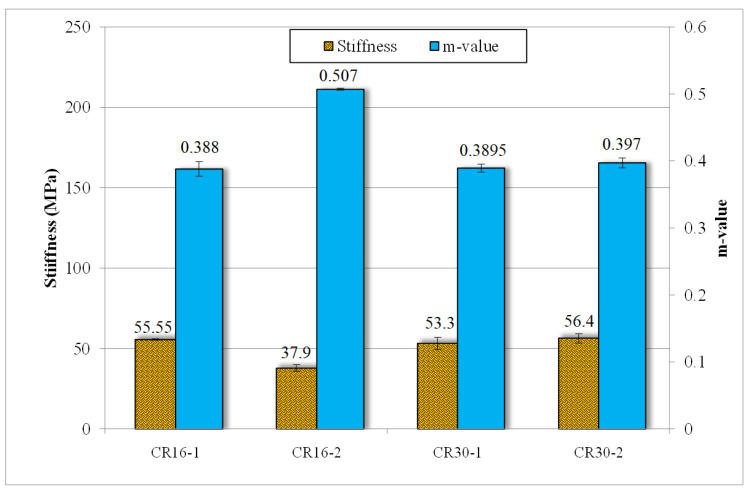
Bending beam rheometer (BBR) results for the rubberized asphalt binders.

**Table 1 polymers-16-01017-t001:** General properties of the PG 64-22 asphalt binder.

Property	Value	Testing Method
Specific gravity @ 15.6 °C	1.041	ASTM D70 [28]
Penetration @ 25 °C	700.1 mm	ASTM D5 [29]
Softening point	46.0 °C	ASTM D36 [30]
Ductility @ 15 °C	>100 cm	ASTM D113 [31]
Cleveland open-cup method flash point	335 °C	ASTM D92 [32]
Mass change after rolling thin-film oven test	−0.013%	ASTM D6 [33]
Absolute viscosity @ 60 °C	179 Pa·s	ASTM D2171 [34]
Stiffness @ −12 °C, 60 s	85.8 MPa	ASTM D6648 [35]

**Table 2 polymers-16-01017-t002:** Experimental conditions for the microbial etching of crumb rubber. CR30 = conditions with 0.6 mm crumb rubber. CR16 = conditions with 1.18 mm crumb rubber. SC = sludge control (no crumb rubber).

Label	Rubber (g)	DI Water (mL)	Mineral Medium (mL)	WAS (mL)	Sulfate (mM)	Thiosulfate (mM)	Acetate (mM)
CR30-1	50	200	0	0	0	0	0
CR30-2	50	0	200	10	0.2	6.3	25
CR30-3	50	0	200	10	0.2	0	25
CR16-1	50	200	0	0	0	0	0
CR16-2	50	0	200	10	0.2	6.3	25
SC	0	0	200	10	0.2	6.3	25

## Data Availability

All data, models, and code generated or used during the study appear in the submitted article.

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
