# Peer review of "Microbially Mediated Rubber Recycling to Facilitate the Valorization of Scrap Tires"

_polymers, 2024, doi:10.3390/polym16071017_

Round 1

Reviewer 1 Report

Comments and Suggestions for Authors

The article deals with rubber desulfurization using tire microorganisms for application in asphalt modification. The article makes important scientific and technological contributions, mainly to sustainability.

Here are some suggestions that can improve the presentation and results of the work:

- would it be possible to analyze the gel content in desulfurized rubber? I understand that the attack will probably be more superficial, perhaps not detectable by soxhlet extraction, but I believe it can give a better idea of the depth of desulfurization.

- what is the origin of the rubber? Tires for passenger cars, trucks, tractors? As the relative amount between natural and synthetic rubber changes depending on the type of application, this could influence the results. If it is a mixed GTR, thermogravimetry could be carried out to at least try to get an idea of the proportion between different types of rubber;

- in item 2.5, please list the physical analyses to be carried out.

- how often were aliquots taken to analyze the amount of sulfate and thiosulfate?

Author Response

The article deals with rubber desulfurization using tire microorganisms for application in asphalt modification. The article makes important scientific and technological contributions, mainly to sustainability.

Here are some suggestions that can improve the presentation and results of the work:

- would it be possible to analyze the gel content in desulfurized rubber? I understand that the attack will probably be more superficial, perhaps not detectable by soxhlet extraction, but I believe it can give a better idea of the depth of desulfurization.

Response to the reviewer: Authors appreciate reviewer’s comments to improve the manuscript. Soxhlet extraction primarily quantifies the extractable (soluble) components of a sample, which may not sensitively reflect the nuanced changes in cross-link density or the presence of superficially altered layers in desulfurized rubber. The method's effectiveness in detecting minor or surface-specific modifications is limited. While further study can be discussed in follow-up research work, this is beyond of the scope of the manuscript.

- what is the origin of the rubber? Tires for passenger cars, trucks, tractors? As the relative amount between natural and synthetic rubber changes depending on the type of application, this could influence the results. If it is a mixed GTR, thermogravimetry could be carried out to at least try to get an idea of the proportion between different types of rubber;

Response to the reviewer: Authors appreciate reviewer’s comments to improve the manuscript. The collected rubber is from passenger car tires. While authors agree that a TGA analyses could give a clearer picture of proper proportion, while collecting from the supplier, it was made sure the source is consistent. So it was assumed that the tire rubbers had similar proportions. Moreover, samples from only one batch of rubber was used for all experiments.

- in item 2.5, please list the physical analyses to be carried out.

Response to the reviewer: Authors appreciate reviewer’s comments to improve the manuscript. The direct physical analysis carried out was Fourier Transform Infrared Spectroscopy (FT-IR), size analysis and rheological analysis which has been discussed in the manuscript.  

- how often were aliquots taken to analyze the amount of sulfate and thiosulfate?

Response to the reviewer: Authors appreciate reviewer’s comments to improve the manuscript. We had 3 samples each.

Reviewer 2 Report

Comments and Suggestions for Authors

The authors have studied the efficiency of a microbially-mediated desulfurization process on rubber particles of different sizes. As a result of this study, the authors conclude that the microbial treatment method was effective regardless of particle size. In general, this paper is a continuation of the number of works in this field made by the authors and other scientific groups. The first question is why the authors used only two sizes of rubber particles (1.18 and 0.6 mm) if the main aim was to investigate the influence of this size on the effectivity of the microbially-mediated desulfurization process?   May be more experimental materials are necessary to make such conclusions?

Some other small moments:

1. Please pay attention: in Abstract and in Conclusions you say about 1.17 mm and 0.595 mm size, but the rest of the text includes 1.18 mm and 0.6 mm. For example, "Conditions labelled “CR30” contained crumb rubber with a diameter of 0.6 mm, while conditions labelled “CR16” contained crumb rubber with a diameter of 1.18 mm (Table 2)." on page 3, lines 95-96. I understand that there is only a small difference, but...

2. page 2, lines 61 and 74. Wrong references ([15] and [23]). It should be corrected.

3. page 4, lines 168-169. "A mid-infrared  range diamond ATR was used to detect changes in xx." What does xx mean?

4. please check the references list for suitability to the Journal rules.

Comments on the Quality of English Language

Minor editing of English language required

Author Response

The authors have studied the efficiency of a microbially-mediated desulfurization process on rubber particles of different sizes. As a result of this study, the authors conclude that the microbial treatment method was effective regardless of particle size. In general, this paper is a continuation of the number of works in this field made by the authors and other scientific groups. The first question is why the authors used only two sizes of rubber particles (1.18 and 0.6 mm) if the main aim was to investigate the influence of this size on the effectivity of the microbially-mediated desulfurization process?   May be more experimental materials are necessary to make such conclusions

Response to the reviewer: Authors appreciate reviewer’s comments to improve the manuscript. The 1.18mm and 0.6mm sizes are two particle sizes that are widely used within the civil engineering applications especially in asphalt binder modification. That is why those were chosen in this study. It is worth mentioning that in a previous study it was successfully demonstrated that asphalt modification with particle sizes <0.25mm has improved performance where key findings include a reduction in disulfide bonds, increased surface energy, and reduced phase separation in the bitumen mixture, contributing to enhanced pavement performance and offering a low-cost, bio-inspired recycling solution for scrap tires [1].

  1. Kabir, S. F., Zheng, R., Delgado, A. G., & Fini, E. H. (2021). Use of microbially desulfurized rubber to produce sustainable rubberized bitumen. Resources, Conservation and Recycling, 164, 105144.

Some other small moments:

  1. Please pay attention: in Abstract and in Conclusions you say about 1.17 mm and 0.595 mm size, but the rest of the text includes 1.18 mm and 0.6 mm. For example, "Conditions labelled “CR30” contained crumb rubber with a diameter of 0.6 mm, while conditions labelled “CR16” contained crumb rubber with a diameter of 1.18 mm (Table 2)." on page 3, lines 95-96. I understand that there is only a small difference, but...

Response to the reviewer: Authors appreciate reviewer’s comments to improve the manuscript. These typos have been corrected in the manuscript.

  1. page 2, lines 61 and 74. Wrong references ([15] and [23]). It should be corrected.

Response to the reviewer: Authors appreciate reviewer’s comments to improve the manuscript. These references have been corrected in the manuscript.

  1. page 4, lines 168-169. "A mid-infrared range diamond ATR was used to detect changes in xx." What does xx mean?

Response to the reviewer: Authors appreciate reviewer’s comments to improve the manuscript. Correction was made so the sentence reads as “A mid-infrared range diamond ATR was used to detect changes in 500 to 540 cm-1 range. “

  1. please check the references list for suitability to the Journal rules.

Response to the reviewer: Authors appreciate reviewer’s comments to improve the manuscript. All references were checked and needed corrections were made.

Reviewer 3 Report

Comments and Suggestions for Authors

Review Report

The manuscript “Microbially-mediated rubber recycling to facilitate valorization of scrap tires refers to the application of recycled scrap tires and problems connected with the recycling of rubbery products and further the use of the recycled material in constructions.

I find this manuscript interesting and worth publishing because it can be the base for the deeper further studies concerning the potential of  microbially assisted desulfurization of rubber.

And the results obtained by authors are interesting due to practical meaning of the proposed rubber recycling method in pavement construction as an environmental friendly and low cost method promoting  reduction in rubber-bitumen separation.    

However there are comments for authors which I would like to point before accepting that work.    

1)     Abstract. It is written concisely.  

2)     Introduction. The introduction is written concisely. In Line 61 reference is written in form “[15]” for all other references the form as “Quin et al, 2020” is used, please correct this. Similar in Line 74 the reference is written in form [23].

3)     Introduction. Could authors in Introduction shortly underline the novelty of this research as compared with the cited literature?

4)     Material and methods This part is written adequately and the chosen research methods are correct. Please correct Line 169, there is “to detect changes in xx”.

5)     Results and discussion  This part is written adequately and the analysis of the data is correct. Please correct in Lines 342-343, 352, 354, 356, 360  the subscripts e.g CO2

Comments on the Quality of English Language

English language fine. No issues detected. Some spelling mistakes to correct as in report for the Authors

Author Response

The manuscript “Microbially-mediated rubber recycling to facilitate valorization of scrap tires refers to the application of recycled scrap tires and problems connected with the recycling of rubbery products and further the use of the recycled material in constructions.

I find this manuscript interesting and worth publishing because it can be the base for the deeper further studies concerning the potential of microbially assisted desulfurization of rubber.

And the results obtained by authors are interesting due to practical meaning of the proposed rubber recycling method in pavement construction as an environmental friendly and low cost method promoting reduction in rubber-bitumen separation.    

However there are comments for authors which I would like to point before accepting that work.    

1)     Abstract. It is written concisely.  

Response to the reviewer: Authors appreciate reviewer’s comments to improve the manuscript. Correction was made.

2)     Introduction. The introduction is written concisely. In Line 61 reference is written in form “[15]” for all other references the form as “Quin et al, 2020” is used, please correct this. Similar in Line 74 the reference is written in form [23].

Response to the reviewer: Authors appreciate reviewer’s comments to improve the manuscript. Correction was made.

Response to the reviewer: Authors appreciate reviewer’s comments to improve the manuscript. These references have been fixed in the manuscript.

3)     Introduction. Could authors in Introduction shortly underline the novelty of this research as compared with the cited literature?

Response to the reviewer: Authors appreciate reviewer’s comments to improve the manuscript. The novelty of the work can be stated as:

“...this current paper studies whether microbial treatment can accommodate rubber particles of various sizes. This in turn can significantly reduce pre-processing time, costs and environmental impacts. Key indicators will include the enrichment of rubber-degrading bacteria, Gordonia and Nocardia, and improved rubber-bitumen interactions.”

Response to the reviewer: Authors appreciate reviewer’s comments to improve the manuscript. These references have been fixed in the manuscript.

4)     Material and methods This part is written adequately and the chosen research methods are correct. Please correct Line 169, there is “to detect changes in xx”.

Response to the reviewer: Authors appreciate reviewer’s comments to improve the manuscript. A mid-infrared  range diamond ATR was used to detect changes in 500 to 540 cm-1 range.

5)     Results and discussion  This part is written adequately and the analysis of the data is correct. Please correct in Lines 342-343, 352, 354, 356, 360  the subscripts e.g CO2

‑Response to the reviewer: Authors appreciate reviewer’s comments to improve the manuscript. These corrections have been fixed in the manuscript.

Round 2

Reviewer 2 Report

Comments and Suggestions for Authors

The Authors have made most of correctioins in the accordance with reviewer's comments. I would like to advice the acceptance of the manuscript for the publication. Before acceptance I kindly ask the Authors to take into account the 4th comment: the reference style and reference list still do not correspond to the journal rules.

E.g. the citation from the Instructions for Authors (Polymers, mdpi journal: https://www.mdpi.com/journal/polymers/instructions):

"In the text, reference numbers should be placed in square brackets [ ], and placed before the punctuation; for example [1], [1–3] or [1,3]. For embedded citations in the text with pagination, use both parentheses and brackets to indicate the reference number and page numbers; for example [5] (p. 10). or [6] (pp. 101–105)."

"References should be described as follows, depending on the type of work:

  • Journal Articles:
    1. Author 1, A.B.; Author 2, C.D. Title of the article. Abbreviated Journal Name YearVolume, page range.
  • Books and Book Chapters:
    2. Author 1, A.; Author 2, B. Book Title, 3rd ed.; Publisher: Publisher Location, Country, Year; pp. 154–196.
    3. Author 1, A.; Author 2, B. Title of the chapter. In Book Title, 2nd ed.; Editor 1, A., Editor 2, B., Eds.; Publisher: Publisher Location, Country, Year; Volume 3, pp. 154–196.   etc...."

Please make the corresponding modifications in the reference numbers in the text as well as the style of the references in the list.

Good luck to the Authors in the future investigations!

Comments on the Quality of English Language

Ok

Author Response

The comment of the reviewer pertaining to placing references in brackets instead of parentheses and adding page numbers is something that the publishing crew can take care of.